# Sustainable Hybrid Lime Mortars for Historic Building Conservation: Incorporating Wood Biomass Ash as a Low-Carbon Secondary Binder

**Jelena Šantek Bajto \*, Nina Štirmer** 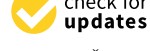 **and Ana Baričević** 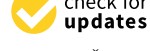

Department of Materials, Faculty of Civil Engineering, University of Zagreb, Fra Andrije Kacica-Miosica 26, 10 000 Zagreb, Croatia; nina.stirmer@grad.unizg.hr (N.Š.); ana.baricevic@grad.unizg.hr (A.B.)
\* Correspondence: jelena.santek.bajto@grad.unizg.hr

**Abstract:** Renewables-based power grid expansion has increased the use of wood biomass as a low-carbon fuel, resulting in the generation of predominantly inorganic wood biomass ash (WBA) as waste during biomass combustion. The conservation of historically valuable, damaged, and energy-inefficient buildings can help downsize carbon emissions and energy consumption, while promoting the use of alternative repair materials, including unavailing materials such as WBA, and implementing zero-waste measures. This study aims to underscore the importance of a proactive approach in managing WBA and its application in artificial hydraulic lime (AHL) mortars. Hybrid lime mortars were prepared by combining natural hydraulic lime (NHL) as the primary binder with fly wood biomass ash (WBA) as the secondary substitute, using different mass ratios of NHL to WBA (100:0, 80:20, and 70:30). The experimental framework encompassed interconnected analytical steps, ranging from binder analysis to paste and mortar preparation. The chemical and mineralogical composition, physical properties, and reactivity of WBA were evaluated to determine the appropriate proportion of WBA for low-carbon AHL binder formulation. Prior to mortar mixing, the water demand, setting time, and soundness of the AHL pastes were assessed. The effects of each AHL binder blend on the mechanical properties of the AHL mortars were analyzed based on compressive and flexural strength measurements after 28 days of curing under different $CO_2$ and moisture conditions ($CO_2$~400 ppm at 70% RH and 95% RH; $CO_2$~30,000 ppm at 60% RH). Additionally, changes in the porous structure were studied. Notwithstanding the greatly prolonged setting time, the results indicate that the mechanical properties of AHL mortars can be enhanced by the addition of WBA in a moderate ratio, empowering the development of environmentally friendly lime mortars suitable for conservation purposes.

**Keywords:** wood biomass ash; hybrid lime binders; low-carbon repair material; enhanced carbonation; sustainable conservation



## 1. Introduction

Following the most devastating earthquakes in the last 140 years, which hit central Croatia in 2020 [1], national plans and strategies narrowed their focus on the comprehensive renovation of buildings, but also on promoting the renovation of buildings with cultural and historical merit, the vulnerability of which was highlighted by the devastation caused by the recent earthquakes. An all-round renovation of these buildings, which are both impaired and energy inefficient, should lead to a reduction in carbon emissions and energy consumption, but also contribute to the development of a circular economy and the application of sustainable solutions. This small niche, which can incorporate other measures to upgrade the basic requirements for construction works already in place, such as using environmentally sound raw and secondary materials in construction [2], can help balance the long-term use of resources while maximizing social benefits and minimizing environmental impacts. In creating a dialogue between the urban environment and

ecosystem resources, the link between sustainable innovation and the cultural dimension is all-important. Just as the European waste hierarchy emphasizes that the zero-waste concept [3] means not merely recycling waste, the maintenance, repair, and renovation of existing buildings must be prioritized over new construction [4]. In this context, effective interventions, particularly in heritage buildings, necessitate extensive knowledge and expertise on the chemical, physical, and mechanical properties of both the existing materials and the materials intended for use [5]. In order to mitigate the damage to the built heritage and, at the same time, the effects of climate change, the selection of suitable repair materials is crucial. In terms of suitability, traditional and contemporary materials are often at odds with each other, making the selection of a suitable, matching material for historic buildings a particular challenge. It is foreseeable that when modern materials are used and applicable codes are followed, heritage preservation often falls by the wayside. In today's world of circular and sustainable architecture, repair materials that are compatible with the materials originally used are therefore of paramount importance. A renewed interest in natural hydraulic lime (NHL) has emerged as it is a useful, relatively modern binder for repair mortars, suitable for the conservation of historic buildings that require very flexible, breathable, yet durable building materials [6,7]. Historical records reveal that hydraulic mortar was anciently made by mixing lime and pozzolans and was used in numerous buildings in ancient Rome and Greece [5,8,9]. Not only does it blend well with authentic building materials, it is also considered a more sustainable alternative to cement [10–12]. This ought to be in line with the European Union (EU) mainstreaming renewable energy sources such as biomass, geothermal resources, sunlight, water, and wind. By following sustainable trends that address complex local conditions and growing energy and material needs, the construction sector is also following the United Nations Sustainable Development Goals (SDGs). The link between the SDGs, focusing on Goals 9, 11, and 12, and the incentive for this research is illustrated by the fact that the use of environmentally friendly technologies, such as biomass, results in additional waste production. Therefore, strategic waste management, i.e., rethinking waste as a resource and preserving built heritage, is essential. This points to the need to include the cultural dimension in a long-term concept of local sustainable development. In order to scale up the manufacturing capacity for the net-zero technologies and products needed to meet ambitious European climate goals [13,14], it is imperative that the EU introduce innovations into mass production that will decarbonize energy-intensive industries such as cement and lime. Currently, lime production is the second largest industrial source of carbon emissions after cement production [15]. Following the European Green Deal, the key EU strategy to address climate change and achieve climate neutrality, the European Commission proposed raising the EU climate target for 2030 to at least a 55 percent reduction in greenhouse gas emissions, which would mean a widespread phase-out of coal by that date [16]. Even more stringent legislation to accelerate the rollout of renewables raises the binding renewable energy target to at least 42.5% by 2030 [17]. Closing the gap left by the decline in coal burning depends largely on the ability of renewables to meet these overall binding targets [18]. According to data of [19], wind, solar, and bioenergy will be the ones providing most of the energy in the EU by 2050. Biomass for energy production is already the most important source of renewable energy in the EU and supports the EU's energy security by holding a share of almost 60% in the total renewable energy mix. Forest biomass is the leading point of biomass supply (logging and wood processing residues, firewood, etc.) [20–22], but can only be considered carbon neutral if it comes from a forest system whose carbon stock is balanced [23]. In comparison, bioenergy consumption in the EU in 2017 was almost 120 Mtoe (58%), while other renewable energy sources such as photovoltaic, wind, and hydropower accounted for about 86 Mtoe (42%) [24]. However, with the increasing number biomass-fired power stations, there is a surplus of waste ash worldwide. It is projected that biomass combustion will generate about 476 million tons of ash annually, assuming that 7 billion tons of biomass are burned with an average ash content of 6.8% [25]. While it may be true that the EU is firmly committed to minimizing

the uncontrolled spread of waste in the environment [26,27], the ongoing WBA handling routine in Europe is grounded on basic landfilling [21]. In the interest of finding pragmatic and sustainable yet value-based functions for WBA more broadly, the vast majority of studies address the beneficial use of WBA in cementitious composites, identifying it as a cement replacement, mineral admixture, or aggregate replacement in concrete. In contrast, research on employing WBA as a pozzolanic and/or hydraulic component in the formulation of hydraulic lime, which is the focus of this research, is still limited. Previous studies have focused almost exclusively on the influence of WBA addition on the properties of air-lime mortars [28–32], while only a few papers in the literature place WBA within the framework of a hydraulic lime framework [33–36]. As hydraulic lime is a favorable material for use in buildings subjected to considerable atmospheric action [37,38], its value must be highlighted when used in historic buildings, instead of air-lime, which is often considered the material most like the materials originally used. The formulation of hydraulic limes with the addition of WBA complies with the EN 459-1 standard [39], which distinguishes between natural and artificial lime and defines artificial hydraulic lime (AHL) (detailed as hydraulic limes (HL) and formulated limes (FL)) as binders containing air lime and/or natural hydraulic lime (NHL) in combination with hydraulic and/or pozzolanic materials. Hydraulic limes vary in their degree of hydraulicity, extended from feebly to eminently hydraulic, but in all of them, two setting mechanisms take place: a hydraulic reaction, i.e., setting with water, and a carbonation reaction, i.e., uptake of $CO_2$, which contributes to the hardening of the material [40,41]. The available lime content, limited in EN 459-1 to FL in the form of $Ca(OH)_2$, widely ranges from 15% to 80% by weight, which means that it is possible to produce artificial hydraulic lime with a high content of artificial hydraulic components [37].

The focus of this work is on the evaluation of AHL binder attributes and their impression on mortar properties as a function of different curing conditions. The results of the study will be beneficial in determining favorable curing conditions and practicable WBA contents for hydraulic lime mortars to optimize their formulation for use in mortars for the conservation and maintenance of historic buildings. Since WBA is an industrial by-product that is intended to replace some of the NHL, it would not only reduce the impact of NHL production on resource depletion and the environment itself, but also solve the problem of industrial waste by cycling resources. Thus, the use of WBA in AHL, especially in higher proportions, could offer an important development perspective by boosting the sustainability of the hydraulic production sector while supporting the authenticity of repair mortars.

## 2. Materials and Methods

### 2.1. Raw Materials and Binder Formulation

Hybrid artificial hydraulic lime (AHL) binders, i.e., binder mixtures of wood WBA and NHL, were prepared with 2 fly WBAs collected from Croatian power plants using untreated wood chips as fuel and operating with grate firing systems. The only source of moderately hydraulic lime used in this study as part of the AHL binary binder system was a commercially available natural hydraulic lime, NHL 3.5, manufactured and supplied by Baumit Croatia. In the power plants themselves, the fly WBA is stored in closed containers so that the ash is not directly exposed to weathering and the influence of moisture and/or the development of carbonation is limited. CEN standard sand, i.e., fine quartz sand with a defined particle size distribution between 0.08 and 2.00 mm, was used as aggregate for all mortar mixtures. Ahead of mortar preparation, AHL was evaluated at the binder and paste levels, with pastes made with binder blends of artificial hydraulic lime (AHL) that differed in the NHL:WBA ratio. The proposed designations for the binder/paste blends are AHLi-20 and AHLi-30, where the "i" stands for the ID number of the WBA instead of 20 and 30 wt.% NHL 3.5. A control lime binder/paste blend (designated AHL0) was prepared using only NHL as the primary binder.

## 2.2. Mortar Mixtures: Preparation and Curing Conditions

The mortars were prepared with binder blends of artificial hydraulic lime (AHL), which differed in the ratio of NHL: WBA and were mixed with potable water and standardized sand. The proposed designations, LMi-20, and LMi-30, for the mortar mixes were chosen to align with the designations for the binder and paste blends. A control lime mortar mix (designated LM0) was prepared using only NHL as the primary binder. Prior to blending the AHL binders, WBA1 was sieved through a 250 μm sieve to remove impurities such as unburned wood and metal pieces or charcoal fragments, which reduced the percentage of impurities by an average of 5%. Since the proportion of impurities in WBA3 was negligible at approx. 1%, this ash was used as collected while preparing the LM3-20 and LM3-30 mortar mixtures. This type of selective removal was intended to overcome physical incompatibilities in the AHL binder system that could result in excessive water requirements or reduced performance [34]. The mass ratio of binder to aggregate was kept constant at 1:3 in all mixes. The water–binder ratio was maintained at 0.60 in all AHL mortar mixes, with the polycarboxylate superplasticizer content adjusted to achieve a flow diameter of 160 to 170 mm, as these flow values indicate the fair workability of mortar on the construction site [42]. An air entraining agent was added to all mortars at a 0.02 mass percent of the binder. The admixtures used in the AHL mortar mixes were supplied by PINKY -S d.o.o., a Croatian manufacturer of admixtures.

A modified mortar mixing procedure was performed according to EN 196-1 [43]. All dry ingredients were mixed and homogenized before water and admixtures were added to simulate a ready-to-use mortar. Standard steel molds ($40 \times 40 \times 160$ mm) were used to cast the mortar specimens. After casting, the specimens were kept in the molds under humid conditions (relative humidity RH of $90 \pm 5\%$ and temperature of $20 \pm 2$ °C) for a period of 3 to 5 days after mixing, according to EN 1015-11 [44]. As soon as demolding was possible, this period being extended by the WBA content, the mortar specimens were pre-cured for up to 7 days under humid conditions and then rearranged for up to 28 days according to the following curing regimen:

(a) humid curing (HC) at a controlled temperature of T = $20 \pm 5$ °C and relative humidity RH = $90 \pm 5\%$, with an average $CO_2$ content between 300 and 400 ppm;

(b) dry curing (DC) under controlled temperature of T = $20 \pm 5$ °C and a relative humidity RH = $60 \pm 10\%$, with an average $CO_2$ content ranging from 300 to 400 ppm;

(c) accelerated curing (ACC) at a controlled temperature of T = $20 \pm 5$ °C and a relative humidity RH = $60 \pm 5\%$, and a $CO_2$ content of 30,000 ppm.

The above curing scheme was established following the report [11], since the degree and sequence of hydration and carbonation reactions are strongly influenced by the moisture content [45,46]. In addition, it is conceivable that the hydration of a pozzolan may also occur under relatively dry curing conditions, depending on the reactivity of the pozzolan itself, implying that mortars would perform better if exposed to an initial moist pre-curing for up to 2 weeks after application and a subsequent lower moisture environment.

Therefore, all AHL mortar mixes were exposed to HC in the early stages to improve the hydration reactions. DC should assist the carbonation reaction, while HC should promote the hydration reactions. ACC was intended to create conditions under which the carbonation reactions are enhanced in a $CO_2$ chamber with a $CO_2$ concentration of 3% by volume and a relative humidity of 60%, because depending on the type and size of the mortar, as well as the atmospheric conditions, carbonation is a process that takes years [47]. The results of excessive $CO_2$ exposure during mortar curing were studied and compared to natural carbonation to mimic mortar exposure conditions in historic buildings. Although it has been found that a higher RH degree allows both higher carbonation and greater hydration of the hydraulic compounds in hydraulic mortars [48–50], the AHL mortar samples were fully carbonated after 28 days and had no visible cracks on the surface. Even though average values between 80 and 90% of final carbonation can be achieved near the surface of mortars exposed to the atmosphere, there is a general consensus in the scientific

literature that a degree of carbonation of 100% is de facto not to be expected in an outdoor environment [51].

*2.3. Methods*

The investigation was carried out at the binder, paste, and mortar level, testing both individual binder components and binder combinations. Prior to the preparation of the mortar mixes, the individual WBAs were evaluated together with the NHL with respect to the physical, chemical, and mineralogical properties as well as the reactivity of the different binder blends (Table 1).

**Table 1.** Test methods for NHL and WBA assessment.

| Level | Property | Test Period | Unit | Standard |
|---|---|---|---|---|
| Binder | Density | Prior to mixing of pastes and mortars | $g/cm^3$ | ASTM C-188-17 |
| | Bulk density | | $kg/dm^3$ | EN 459-2:2021 |
| | Free water | | wt.% | |
| | Particle size distribution | | % | EN 459-2:2021 (Air-jet sieving) |
| | | | μm | Laser diffraction method (ISO 13320:2020) |
| | Chemical composition | | wt.% | ISO/TS 16996:2015 |
| | Loss on ignition (LOI) | | | ASTM D 7348-13 |
| | pH value | | - | EN ISO 10523:2005 |
| | Heavy metal concentrations | | mg/kg | ISO/TS 16996:2015 |
| | Phase/composition identification | | wt.% | Thermogravimetric analysis (TGA) |
| | | | - | X-ray diffraction analysis (XRD) |

In addition, the contribution of the individual AHL binder blends to the properties of the pastes was analyzed using the test methods shown in Table 2.

**Table 2.** Test methods for AHL paste assessment.

| Level | Property | Test Period | Unit | Standard |
|---|---|---|---|---|
| Paste | Standard consistency | Immediately upon mixing of pastes | % | EN 459-2:2021 |
| | Setting time | | h | |
| | Soundness | | mm | |
| | Reactivity (Hydration kinetics) | | W/g | EN 196-11:2019 |

Thermogravimetric analyses (TGA) of NHL and WBAs were performed on a Discovery TGA 55 from TA Instruments in a temperature range between 30 °C and 1000 °C at a constant heating rate of 10 °C/min using inert gas nitrogen at a flow rate of 40 mL/min.

Powdered samples of WBA and NHL were subjected to qualitative analysis using X-ray diffraction (XRD). The X-ray diffraction data were collected at room temperature using a Bruker D8 Discover diffractometer equipped with a LYNXEYE XE-T detector, following Bragg–Brentano geometry. The XRD data were collected over a 2θ range of 20–80° using a step size of 0.02°, and each data point was recorded for 25 s. Copper (Cu) radiation (K-Alpha1: 1.54060 Å, K-Alpha2: 1.54443 Å) was employed, while the generator operated at a constant 40 mA and 40 kV configuration. The main focus of the analysis was on qualitative assessment, which involved identifying the presence or absence of specific components without quantifying their precise concentrations. The analysis involved a semi-quantitative Rietveld refinement using the HighScore Xpert Plus program 3.0, despite the qualitative presentation of the results. The refinement process incorporated the split-type pseudo-Voigt profile function and the polynomial background model. It assumed isotropic vibration modes for all atoms. During the refinement, parameters including zero shift, scale factor, half-width (W), asymmetry (if applicable), and peak shape were simultaneously adjusted. Notably, the analysis did not refine the atomic coordinates.

Pulverized slices of fully carbonated and non-carbonated mortar samples were also analyzed using XRD to characterize the mineralogical composition at 28 days of age.

Calorimetric analysis of the AHL binder mixtures, i.e., measurement of the heat of hydration of the mixed pastes, was performed using an 8-channel isothermal TAM Air 8-Channel Standard Volume Calorimeter according to EN 196-11:2019 [52]. The heat of hydration was measured for 3 days at 20 °C, and a w/b ratio of 0.6 was chosen to achieve a paste consistency suitable for filling the ampoules. Approximately 30 g of the paste was mixed manually for 2 min, and 10 g of the prepared sample was then placed in a sealed ampoule and lowered into the calorimeter conditioned to 20 ± 0.05 °C.

Furthermore, the contribution of the individual binder compositions to the mortar properties in both fresh and hardened state was analyzed using the test methods and recommendations listed in Table 3.

**Table 3.** Test methods for assessment of AHL mortar.

| Level | Property | Test Period | Unit | Standard |
|---|---|---|---|---|
| Fresh mortar | Bulk density | Immediately upon mixing of mortars | kg/m$^3$ | EN 1015-6:2000 |
| | Temperature | | °C | EN 12350-1:2019 |
| | Air content | | % | EN 459-2:2021 |
| | Consistency of fresh mortar (by flow table) | | mm | |
| Hardened mortar | Compressive strength | After 28 days of different curing regimes | MPa | EN 1015-11:2019 |
| | Flexural strength | | | |
| | Pozzolanic reactivity | | % | ASTM C618-22 |
| | Carbonation degree | | mm | 'Repair Mortars for Historic Masonry' [45] |
| | Phase/composition identification | | - | X-ray diffraction (XRD) analysis |
| | | | | Scanning electron microscopy (SEM) |
| | Porosity | | % | Mercury intrusion porosimetry (MIP) |

The pozzolanic reactivity of the WBA was evaluated using ASTM C618, with a strength activity test as a performance indicator. As per the guidelines outlined in ASTM C618, pozzolanic reactivity is interpreted when the activity index (SAI) exceeds 75% after 28 days of testing. Specifically, when 20 wt.% of the binder is substituted with coal fly ash or natural pozzolans, achieving an SAI greater than 75% confirms the desired pozzolanic reactivity. The strength activity index was expressed as the ratio between the compressive strength of the mortar with partial WBA substitution and the compressive strength of the reference mortar prepared with only NHL as the primary binder. The compressive strength alone was measured on the halves of three prismatic specimens in a measuring range of $400 \pm 40$ N/s.

To observe the degree of carbonation, mortar specimens exposed to the curing conditions described above (HC, DC, ACC) for 28 days were divided in half and then a 1% phenolphthalein alcohol solution was sprayed on the corresponding section. The carbonation depth was measured on each side of the divided mortar half using a precise electronic measuring device, and the carbonation depth was then calculated as an average value [53].

Scanning electron microscopy (SEM) analysis of the cured mortar samples was performed using the JSM-IT200 microscope from JEOL. Prior to analysis, the samples underwent a gold coating procedure to improve image contrast. The SEM analysis was conducted using an acceleration voltage of 15 kV, and a consistent probe current of 50 nA was maintained across all samples. This technique is especially effective for untreated samples, as it enhances the visibility of slopes, pores, and other morphological features. The analysis employed a secondary electron detector (SED), chosen for its ability to provide high-resolution surface images. This detector proved to be ideal for studying the intricate morphology of the samples.

Mercury intrusion porosimetry (MIP) was used to measure the pore structure parameters of fully carbonated and non-carbonated mortar samples. The pore entry diameter was calculated by combining the results of Micromeritics' AutoPore IV 9500, assuming a contact angle of 130° and a maximum pressure of 420 MPa.

## 3. Results and Discussion

### 3.1. Binder Assessment

3.1.1. Physical Properties

As part of the basic physical characterization of NHL and WBAs, specific gravity, bulk density, residue on the 90 μm and 120 μm sieve, free water content, and particle size distribution were evaluated. On visual inspection, the WBAs used in this work did not exhibit any apparent characteristics compared to the NHL other than the difference in coloration. While the NHL had a lighter, yellowish-beige color, the WBA1 tended to be greyer and the WBA3 was taupe in color (Figure 1). None of the ashes appeared to have impurities visible to the naked eye, as determined by sieving.

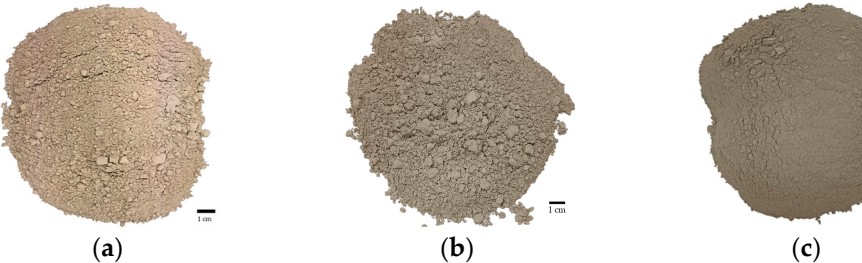

| (a) | (b) | (c) |

**Figure 1.** Visual features of (**a**) NHL 3.5, (**b**) WBA1, and (**c**) WBA3.

The lowest specific gravity value is reported for WBA1 (2.66 g/cm$^3$), but all three values were quite comparable (Table 4). Bulk density ranged from 0.288 (WBA1) to 0.413 (WBA3) and differed greatly from the highest value of 0.789 kg/dm$^3$ in NHL.

**Table 4.** Physical properties of NHL and WBAs vs. limits set for NHL and FL.

| NHL/WBA ID | NHL 3.5 | WBA1 | WBA3 | EN 459-1 Criteria for NHL/FL |
|---|---|---|---|---|
| Specific gravity (g/cm$^3$) | 2.68 | 2.66 | 2.69 | - |
| Bulk density (kg/dm$^3$) | 0.789 | 0.288 | 0.413 | - |
| Residue on 90 μm (%) | 13.1 | 0.8 | 11.4 | $\leq$15 |
| Residue on 120 μm (%) | 2.0 | 0.3 | 0.4 | $\leq$5 |
| Free water (%) | 0.2 | 0 | 0 | $\leq$2 |

When determining the particle size of the fine particles by air-jet sieving, both WBA1 and WBA3 met the criteria of EN 459-1, which states that the residue on the 90 μm and 120 μm sieves must be less than or equal to 15% and 5% of the binder mass, respectively. The residue on the 90 and 120 μm sieve was highest for NHL 3.5, followed by WBA3 and WBA1. Since this method is considered unsuitable for materials with fine fractions such as WBA, whose fine particles tend to stick together and break up strongly during the sieving process [54], these results are supported by laser diffraction analysis (Figure 2).

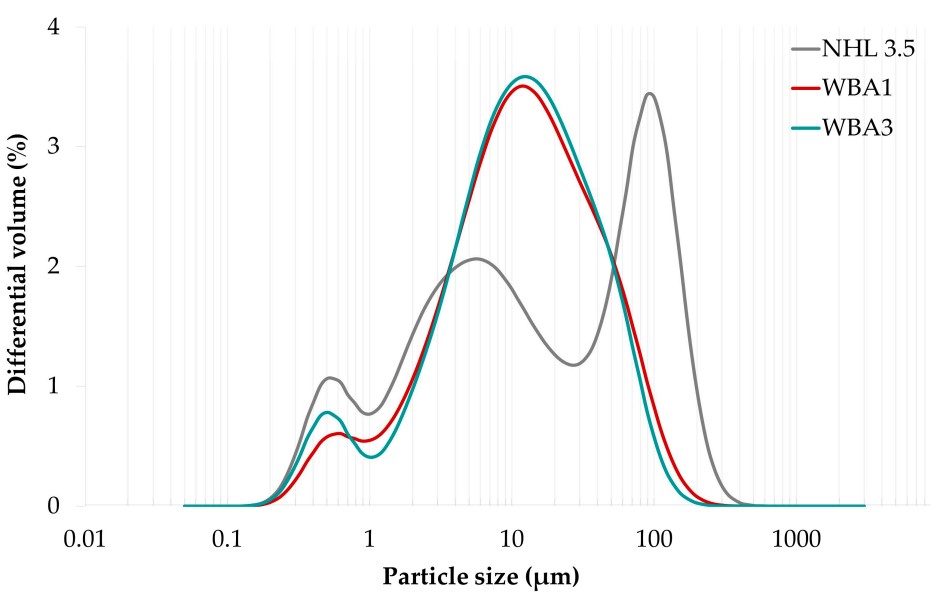

**Figure 2.** Differential volume distribution of particle sizes in NHL 3.5 and WBAs.

Laser diffraction analysis of binder particle size was performed using a Shimadzu SALD-3101 analyzer. Samples were dispersed in an air stream at a pressure of 0.4 MPa, and results were analyzed using the Fraunhofer approximation. Particle sizes of both ashes ranging from 0.1345–50 μm accounted for 90 wt.% of the total ash mass. The median particle size (D50) for both WBAs was 10.97 μm, with the WBAs characterized by a bimodal particle size distribution with a dominant region of about 12 μm in diameter, which is not visible in the NHL sample. Both NHL and WBAs exhibit a similar ultrafine region with a diameter of about 0.5 μm. In addition, the NHL is characterized by a trimodal particle size distribution that includes the smallest ultrafine region with a diameter of about 0.5 μm, a fine fragmentation region with a diameter of about 6 μm, and a large fragmentation region for particles with a diameter of about 90 μm (Figure 2). The particle sizes of NHL 3.5 ranged up to 123 μm, accounting for 90% by weight of the total mass, while the value of the median particle size (D50) was 13.67 μm.

While the particle size distribution of NHL 3.5 shows polydispersity, the distribution for both WBAs is narrower and shifted to the left, suggesting a greater proportion of

particles in the fine fragmentation region. The particle size data, i.e., D10, D50, and D90, corresponding to the percentages of 10%, 50%, and 90% of particles below the specified particle size, confirm a broader distribution of particles in the NHL, as well as similar values for both sieved (WBA1) and unsieved (WBA3) ash (Table 5). In general, fly ash particles between 10 and 45 μm are thought to be responsible for the strength increase starting at 28 days and lasting up to about one year [55]. This particle size range contains about 40% of the particles of both ashes, while the cumulative value of a matched volume in the NHL sample is reached only at 90 μm, indicating a somewhat higher fineness of the WBAs than of the NHL.

**Table 5.** Particle size data (D10, D50, and D90) for NHL and WBA samples.

|  | D10 (μm) | D50 (μm) | D90 (μm) |
|---|---|---|---|
| NHL 3.5 | 0.87 | 13.67 | 123.44 |
| WBA1 | 1.69 | 10.97 | 51.19 |
| WBA3 | 1.69 | 10.97 | 45.86 |

### 3.1.2. Chemical and Mineralogical Properties

The chemical composition of NHL 3.5 and WBAs was studied using energy dispersive X-ray fluorescence spectrometry (EDXRF), which allows both qualitative and quantitative analyses. The resulting major oxide data are compiled and listed in Table 6. In addition to the quantitative XRF analysis, Table 6 also presents qualitative XRD results, which offer additional insights into the crystal structure and composition of the samples. The qualitative findings in Table 6 describe the presence of minerals, ranging from minor (less than 5%) to massive (more than 30%). A star (*) is used as a footnote marker in the table to indicate supplementary information. The corresponding footnote, located at the bottom of the table, provides further details on the presence of these minerals. The EDXRF was also used to analyze the heavy metal concentrations in the homogenized dry materials (Table 7).

**Table 6.** Chemical and mineralogical composition of NHL 3.5 and WBAs.

| | Chemical Composition | | | | Mineralogical Composition | | |
|---|---|---|---|---|---|---|---|
| NHL/WBA ID | NHL 3.5 | WBA1 | WBA3 | | NHL 3.5 | WBA1 | WBA3 |
| Element | Percentage [wt.%] | | | Phase | Presence * | | |
| $SiO_2$ | 35.06 | 24.36 | 18.43 | calcite | ++++ | ++++ | ++++ |
| CaO | 52.81 | 35.42 | 41.90 | portlandite | +++ | + | + |
| $SO_3$ | 0.73 | 6.28 | 11.49 | quartz | ++ | ++ | - |
| $Al_2O_3$ | 5.39 | 5.64 | 3.32 | hatrurite, monoclinic/M3-alite | ++ | - | - |
| $Fe_2O_3$ | 2.66 | 2.60 | 1.63 | hatrurite/alite | ++ | + | - |
| MgO | 1.97 | 4.60 | 4.88 | gehlenite | + | ++ | ++ |
| $P_2O_5$ | <0.01 | 4.56 | 4.81 | calcium oxide | - | +++ | +++ |
| $Na_2O$ | 0.55 | 0.89 | 1.03 | periclase | - | + | ++ |
| $K_2O$ | 0.51 | 14.87 | 11.90 | calcio olivine | - | ++ | +++ |
| $TiO_2$ | 0.09 | 0.24 | 0.16 | larnite/ β-belite | +++ | - | - |
| MnO | 0.25 | 0.58 | 0.46 | dolomite | - | + | - |
| | | | | | + | minor presence | |
| | | | | | ++ | moderate presence | |
| LOI | 16.10 | 16.60 | 8.30 | | +++ | major presence | |
| | | | | | ++++ | massive presence | |
| | | | | | - | not present | |

**Table 7.** Heavy metal content in NHL 3.5 and WBAs.

| Heavy Metal (mg/kg) | NHL/WBA ID | | |
|---|---|---|---|
| | NHL 3.5 | WBA1 | WBA3 |
| Zn | <1.0 | 2590.0 | 3520.0 |
| Pb | 1.5 | 10.5 | 20.4 |
| Cu | <1.0 | 95.0 | 64.2 |
| Ba | 84.5 | 1000.5 | 1850.0 |
| Cd | <1.0 | 6.9 | 23.6 |
| Cr | 12.9 | 21.9 | 12.2 |
| Hg | 0.038 | 0.168 | <0.005 |
| Mn | 1755.0 | 5015.0 | 3885.0 |
| Ni | <1.0 | <1.0 | 4.3 |
| Bi | <1.0 | <1.0 | <1.0 |
| Sr | 344.0 | 339.0 | 698.5 |

In the NHL sample, calcium oxide ($CaO$) emerges as the predominant oxide, with $SiO_2$ content closely following. Similarly, in the WBA samples, $CaO$ is also the predominant oxide, with $SiO_2$ content as a close second, aligning with the characteristic composition found in wood materials [56]. NHL 3.5 is mainly composed of four oxides: $CaO$ (52.81%), $SiO_2$ (35.06%), $Al_2O_3$ (5.39%), and $Fe_2O_3$ (2.66%). Although WBA1 is more similar to NHL in terms of these predominant oxides, the $SO_3$, $MgO$, $P_2O_5$, and $K_2O$ values stand out significantly in both WBA1 and WBA3 compared to the NHL sample. Looking at the $SO_3$ content, both WBAs are fly ashes high in $SO_3$, and the measured $SO_3$ content in both WBAs is well above the 2% maximum prescribed by EN 459-1 for formulated lime, but this is somewhat offset by the weighted average value when the ratio of NHL to WBA is 70:30.

Both WBAs were found to contain very high levels of $K_2O$ and $MgO$, indicating that they are highly enriched in alkaline-earth and alkaline compounds, consistent with previous studies by Vassilev et al. [57]. In view of the assumption that the alkalis in the WBA serve as activators for the pozzolanic activity of the WBA with the $Ca(OH)_2$ present in the NHL and subsequently combine with calcium silicate hydrates in the long term [58], higher ash additions were also investigated in this study.

The multi-component, heterogeneous, and non-uniform structure of WBA consists pre-dominantly of inorganic compounds, i.e., amorphous, crystalline to semi-crystalline minerals, and a negligible amount of organic compounds [57,59].

The X-ray diffraction (XRD) analysis findings, as outlined in Table 6, unequivocally demonstrate that calcite ($CaCO_3$) is the prevailing mineral phase in both NHL 3.5 and WBAs. Notably, the NHL sample exhibits significant proportions of portlandite and β-belite (larnite), whereas these phases are either present in minor quantities or entirely absent in the WBA samples. Quartz and hatrurite, manifesting as both monoclinic and hexagonal structures (M3-alite and alite, respectively), exhibit a moderate presence in NHL. Conversely, the WBA samples display a moderate presence of quartz, a minor presence of alite, and a complete absence of β-belite. Gehlenite, a calcium aluminosilicate commonly encountered in natural hydraulic lime [60], manifests as a moderate presence in both WBAs, albeit in lesser amounts in NHL.

The XRF analysis corroborates that calcium oxide exhibits a major presence in both WBA1 and WBA3. Among the moderately present minerals, periclase is discernible in WBA3, with a lower occurrence in WBA1. The abundance of calcio olivine ($Ca_2SiO_4$) is evident in both ash samples, whereas dolomite is detected in minor quantities in WBA1.

Remarkably, the XRF analysis showcases a notable elevation in the levels of $SO_3$ and $K_2O$ within the WBAs. This compelling observation substantiates the plausibility of asserting that the presence of corresponding mineral phases in carbonate-rich **WBA** samples might be veiled under the prevailing abundance of calcite. The discernible outcome of the

XRD analysis highlights an intriguing overlap between the characteristic peaks generated by calcite and those originating from crucial minerals, including sulfides, sulfates, and potassium-bearing minerals. Consequently, this intricate amalgamation of peaks hampered the accurate identification of these minerals during XRD analysis, thus accentuating the formidable challenge imposed by the pervasive presence of calcite [61].

To approximate the hydraulicity of a binder mixture, the weighted average of the elements in AHL binder mixtures is given in Table 8, taking into account the content of each **WBA** in the binder mixture. In appraising the weighted averages, each value from the data set shown in Table 6 was multiplied by a predetermined weighting factor corresponding to the proportion of **NHL** and **WBA**, as follows:

$$\text{X } weighted = \sum \text{X}_{NHL} \times \text{weighting factor (NHL)} + \text{X}_{WBA} \times \text{weighting factor (WBA)} \quad (1)$$

**Table 8.** The weighted average of elements in AHL binder blends.

| Binder Mix ID | AHL0 | AHL1-20 | AHL1-30 | AHL3-20 | AHL3-30 |
|---|---|---|---|---|---|
| Element | \multicolumn Weighted Average [wt.%] | | | | |
| $SiO_2$ | 35.06 | 32.92 | 31.85 | 31.73 | 30.07 |
| CaO | 52.81 | 49.33 | 47.59 | 50.63 | 49.54 |
| $SO_3$ | 0.73 | 1.84 | 2.40 | 2.88 | 3.96 |
| $Al_2O_3$ | 5.39 | 5.44 | 5.47 | 4.98 | 4.77 |
| $Fe_2O_3$ | 2.66 | 2.65 | 2.64 | 2.45 | 2.35 |
| MgO | 1.97 | 2.50 | 2.76 | 2.55 | 2.84 |
| $P_2O_5$ | <0.01 | 0.91 | 1.37 | 0.96 | 1.44 |
| $Na_2O$ | 0.55 | 0.62 | 0.65 | 0.65 | 0.69 |
| $K_2O$ | 0.51 | 3.38 | 4.82 | 2.79 | 3.93 |
| $TiO_2$ | 0.09 | 0.12 | 0.14 | 0.10 | 0.11 |
| MnO | 0.25 | 0.32 | 0.35 | 0.29 | 0.31 |
| $Na_2O_{eq}$ | 0.89 | 2.84 | 3.82 | 2.48 | 3.28 |
| HI | 0.79 | 0.79 | 0.79 | 0.74 | 0.71 |
| CI | 1.91 | 1.89 | 1.89 | 1.77 | 1.70 |

As noted by Gualtieri et al. [62], chemical data alone are not sufficient to accurately characterize the hydraulic properties of hydraulic limes. However, the cementation index (**CI**), one of the most commonly used methods to evaluate the hydraulic properties of lime, has been used to predict the hydraulic properties of the AHL binders [63]. The **CI** was originally developed by Eckel [64] and is calculated based on the weight contribution of different oxides (Equation (2)). As the **CI** increases, it is anticipated that the hydraulic properties of the binder will improve. The same analogy applies to the classification of lime according to the hydraulic index (**HI**) developed by Taylor [65], which balances the most active oxides (Equation (3)).

$$\text{CI} = \frac{1.1 \times Al_2O_3 + 0.7 \times Fe_2O_3 + 2.8 \times SiO_2}{CaO + 1.4 \times MgO} \quad (2)$$

$$\text{HI} = \frac{Al_2O_3 + Fe_2O_3 + SiO_2}{CaO + MgO} \quad (3)$$

The amount of **CaO** is similar in all AHL binder mixtures. AHL1-30 has a higher amount of alumina, magnesium, and iron oxides. AHL3-30, on the other hand, has a higher amount of magnesium oxide. When comparing AHL3-30 with AHL1-30 and AHL1-20 as well as AHL3-20, AHL3-30 has a lower content of silica, alumina, and iron oxides, which mainly give the binders their hydraulic properties [66]. From this analysis, AHL1 binder mixtures are more likely to be hydraulically reactive compared to AHL3. This is also

noticeable when comparing the **CI** and **HI** values, which are similar and higher for AHL1 than for AHL3, while they have similar values to **NHL** 3.5. Nevertheless, both the **CI** and **HI** values, which are above the value of 0.7 as the lower limit for eminently hydraulic limes [65,67] for all investigated AHL mixtures, indicate highly hydraulic materials, akin to natural cement.

Prior to blending the AHL binders, TGA was performed on unmixed powders of NHL 3.5 and WBAs. Approximately 50 mg of dry material was heated at a rate of 10 °C/min in the range of 30–1000 °C in an inert He atmosphere at a flow rate of 40 mL/min (see Figure 3). In general, the thermal behavior of WBA is driven by its chemical composition, which is controlled by the fuel used, the type of boiler, the operating conditions, and the combustion technology. The main constituents of WBA formed in biomass boilers at typical operating temperatures are calcium oxide (CaO), silicon dioxide ($SiO_2$), calcite ($CaCO_3$), portlandite ($Ca(OH)_2$), and calcium silicate ($Ca_2SiO_4$) [68], comparable to the WBAs investigated in this study.

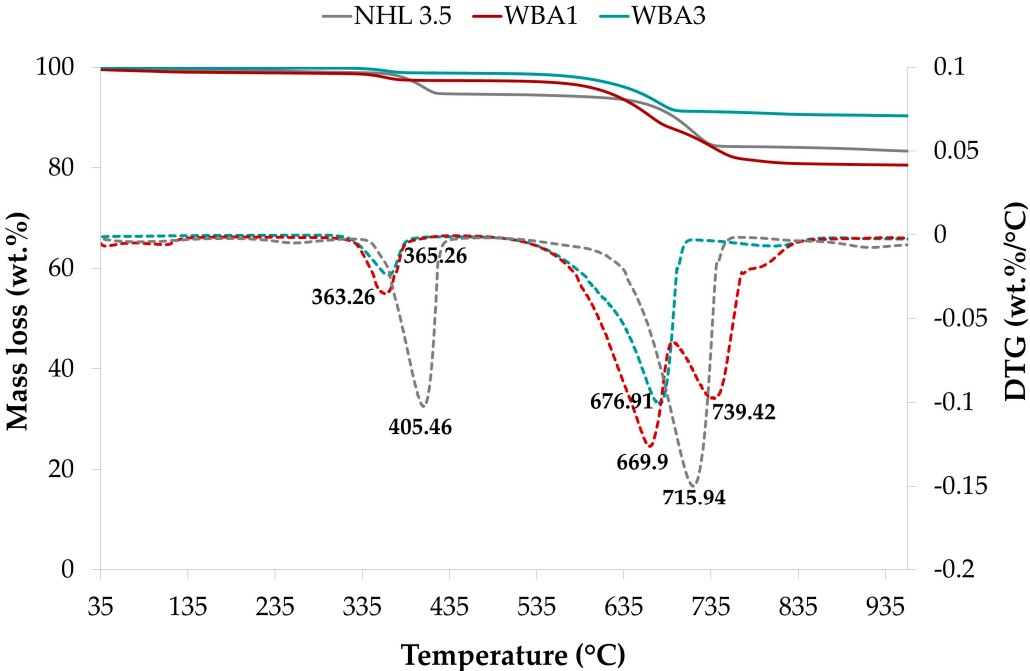

**Figure 3.** TG and DTG curves of powdered NHL 3.5 and WBAs.

No significant mass losses were observed in the interval 50–200 °C, which corresponds to the evaporation of the moisture present, since no free water was exposed. This resulted in two predominant phases of weight loss in both NHL and WBA. The first predominant range of mass loss from 300 °C to 500 °C can be attributed, at least in part, to the decomposition of portlandite, which is the only one of the aforementioned minerals present in the NHL and WBA that is expected to dehydroxylate in this temperature range [68]. The available lime in NHL and WBA, i.e., the $Ca(OH)_2$ content, was determined based on mass loss in the 300–500 °C range, while the calcite content present in the samples was determined based on the mass loss in the second dominant range of 550–850 °C, which corresponds to $CO_2$ release from carbonates [68–70]. When comparing the DTG curves of WBAs and NHL, it is noticeable that the endothermic peaks in both regions are shifted to the left, i.e., the decomposition temperature is somewhat lower. Furthermore, it is noteworthy to highlight that among the samples, WBA1 stands out as the sole specimen displaying two distinct peaks in the decarbonation zone. This phenomenon can be attributed to the presence of dolomite within this specific sample [71].

The content of calcium carbonate and portlandite calculated from the TGA measurements according to the method described in [71] is shown in Table 9 and compared with the qualitative results of the XRD analysis.

**Table 9.** Comparison of carbonates and portlandite content from TG and XRD measurements.

| NHL/WBA ID | CaCO₃ (%) | | Ca(OH)₂ (%) | |
|---|---|---|---|---|
| | **XRD** | **TG** | **XRD** | **TGA** |
| NHL 3.5 | | 23.7 | major presence | 18.3 |
| WBA1 | massive presence | 36.9 | minor | 6.1 |
| WBA3 | | 18.0 | presence | 4.4 |

The content of carbonates and portlandite from the TG measurements agrees with the XRD findings. Based on the TG measurement results, it appears that NHL 3.5 falls short of meeting the specifications outlined in EN 459-1, which necessitates a minimum of 25% lime availability in the form of $Ca(OH)_2$. It is worth noting that over time, the samples may have undergone aging, leading to the stabilization of the water-based additives (WBAs) as well as NHL through the process known as carbonation.

*3.2. AHL Paste Assessment*

3.2.1. Hydration Kinetics

The hydration kinetics, i.e., the evolution of the specific hydration power along with the hydration energy over 72 h, is shown in Figure 4, which plots the heat evolution power in AHL binder blends containing 20 and 30 wt.% WBA with respect to the reference AHL blend containing 100% NHL 3.5. The flow rate has been normalized to 1 g of NHL to indicate reactivity, while the total amount of heat is shown normalized to the initial water volume (J/mL of water), as it has been shown that this normalization can be correlated with compressive strength [72]. All curves showed relatively high initial values (marker 1), which can be attributed to the heat of wetting and the initial dissolution of the available lime [73]. The initial values of hydration power were significantly higher for the AHL1 and AHL3 binder mixtures than for the reference mixture AHL0, with the mixture containing the highest WBA content of 30 wt.% (AHL3-30) showing the highest value of all. After about 16 h, the lines of AHL1-30 and AHL3-30 run almost parallel, with AHL1-30 outperforming the initially more reactive AHL3-30 blend (following about 30 h). While the AHL1-30 reaction process is initially slower than AHL3-30, after 12 h, it demonstrates a faster and more energetic reaction, which can be attributed to the hydration of the belite phase [67]. Binder mixtures AHL3-20 and AHL3-30 show thermal peaks between 20 (marker 4) and 29 h (marker 3). At the same time, AHL1-20 showed a thermal peak at 20 h (marker 2), which was higher than that of AHL3-20. After about 36 h, all mixtures with WBA show a decreasing trend. The specific hydration energy, where the slope of the curve is related to the reaction rate, shows that all binders continue to exhibit hydraulic reactions until the end of the calorimetric test (72 h).

The reference mixture AHL0 exhibited the lowest specific hydration power of all pastes tested and showed a continuous downward trend over the entire test period. The absence of thermal peaks in the AHL0 binder, but also their secondary presence in the AHL1 and AHL3 binders, is due to differences in hydraulicity. Despite the moderately high total $SiO_2$ content, which is expected to bring forth the hydraulic capacity of the binders through the formation of belite polymorphs ($C_2S$), the hydraulic substance in the AHL0 binders did not appear to be sufficient to trigger detectable hydration processes in the indicated test time, consistent with the slow reaction of dicalcium silicate ($C_2S$).

AHL1-30 and AHL3-20 initially exhibit similar reaction kinetics (up to 5 h) despite the difference in WBA content. As mentioned earlier, NHL and the two WBAs have similar fineness, with WBA3 having a slightly higher content of ultrafine particles in this range, with a diameter of about 0.5 μm, and therefore being potentially more reactive at the

beginning of the calorimetric testing. In view of the absence of noticeable exothermic peaks, the reference paste AHL0 (100% NHL 3.5) has a relatively low magnitude and approximately the same stance; it can be concluded that the NHL hydration has no direct influence on the heat power from 5 h onwards, which is consistent with the literature data [74].

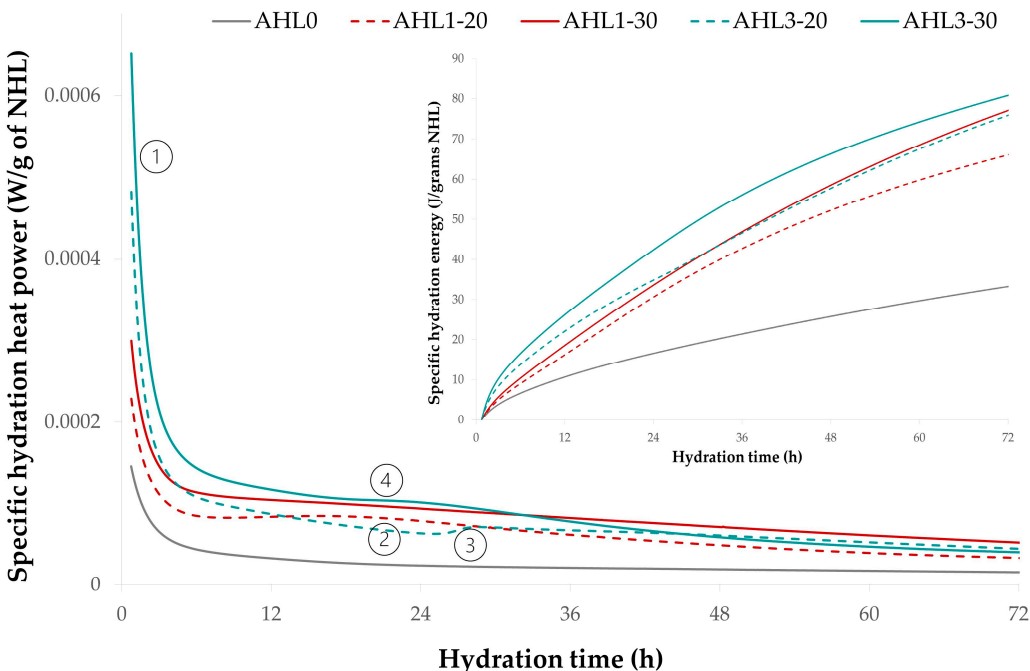

**Figure 4.** Specific hydration heat power in AHL binder mixtures, normalized per 1 g of NHL 3.5.

### 3.2.2. Setting Time, Standard Consistency, and Soundness

As initial setting is considered to be closely related to hydration kinetics [72], the setting time of AHL pastes was analyzed by observing the penetration of a needle into AHL pastes of standard consistency, as specified in EN 196-3, that have a certain resistance to penetration by a standard plunger. The water required to achieve such a paste consistency was determined by trial penetrations into pastes of varying water content. The initial and final set was then measured using a ToniSET automatic Vicat needle instrument. Even though the results of isothermal calorimetry and the setting times measured by Vicat are not fully comparable due to the different amounts of water, some indications can be derived. It is noticeable that all mixtures with WBA are slightly shifted to the right and show higher values of normalized specific hydration power than the AHL0 mix; therefore, one could expect a delay in setting as well as higher values of compressive strength.

In accordance with EN 459-2, an alternative method was used to test the soundness of AHL pastes. After mixing 75 g of the AHL binder with 20 mL of water, three test specimens were prepared (per binder mixture) by immediately filling Le Chatelier molds and placing them in a steam cabinet for 180 min. After this time and allowing the Le Chatelier molds to cool to room temperature, the distance between the ends of the indicator points was measured.

The effect of the WBA on the above properties of AHL pastes was clearly seen in the increase in water requirement to achieve standard consistency, in the prolongation of initial and final setting time, and in volume expansion (Table 10). In determining the amount of water required for a paste of standard consistency, it was found that the AHL binder blends are very sensitive to even slight changes in water addition. As a result, all AHL mixtures showed the same behavioral trend of increased water demand, which increased with the amount of WBA used. Ruggedly shaped and highly porous particles of fly WBA, which are assigned a larger specific surface area, as confirmed by previous

WBA characterizations [18,75,76], may be responsible for the increased demand for water. Although the increased water requirement was coupled with extended setting, a divergence between the two is highlighted as both the initial and final set were extremely delayed for AHL pastes with WBA while the maximum increase in water demand was at most 5% (AHL1-30). The AHL3-30 paste did not meet the criteria of EN 459-2 as the final setting time occurred after 42 h, surpassing the final setting time limit of 30 h. For the other three mixtures (AHL1-20, AHL1-30, and AHL3-20), the final setting was completed before the 30 h limit, although the period was up to three times longer than the final set of the reference mixture. The fierce hindering effect of WBA on the setting of hydraulic binders was attributed by [72,76–78] to the high contents of sulfates and heavy metals, especially zinc (Zn), lead (Pb), and copper (Cu), that are usually present in WBA. The predominance of the $SO_3$ and Zn content (see Tables 6 and 7), as well as a very high content of other heavy metals, was also confirmed in the WBAs tested in this study; for example, the Zn content in WBA3 was 3520.0 mg/kg, while it was absent in the NHL sample. In addition, the highest $SO_3$ content of 3.96% was found in AHL3-30 (see Table 9), the paste that also exhibited the longest setting time of 42 h. In addition, a high $P_2O_5$ content could also lead to a delay in setting, as could a high alkali concentration, by preventing $Ca^{2+}$ dissolution [79,80]. Both anomalies, elevated phosphate, and very high alkali content, were also observed in the two WBAs used. Thus, the retarded setting of the AHL pastes is strongly influenced by the chemical composition of the ashes.

**Table 10.** Influence of WBA on the properties of AHL pastes.

| AHL ID | | AHL0 | AHL1-20 | AHL1-30 | AHL3-20 | AHL3-30 | EN 459-1 Criteria |
|---|---|---|---|---|---|---|---|
| Water requirement for standard consistency (%) | | 40 | 44 | 45 | 41 | 42 | - |
| Setting time (h) | Initial set | 5 | 7.5 | 13 | 15 | 21 | >1 |
| | Final set | 10 | 20 | 25 | 29 | 42 | <30 |
| Soundness (mm) | | 5.22 | 7.39 | 9.23 | 6.64 | 8.70 | ≤20 |

The average soundness values of 7.02 and 8.97 mm obtained for AHL mixes at 20 and 30 wt.%, respectively, and the maximum value of 9.23 mm for AHL1-30 are well below the maximum allowable soundness value of 20 mm given in EN 459-1 [39]. This is an indication that inclusion of wood ash up to 30% as a substitute for NHL does not result in unsound mortar. However, the elevated values, i.e., the increase in expansibility of the prepared AHL pastes, could be related to the high content of periclase (MgO) [81], which appears in the chemical and mineralogical analysis of the WBAs.

*3.3. AHL Mortar Assessment*

3.3.1. Fresh-State Properties

The determination of the consistency of fresh AHL mortars is used to assess their workability in order to obtain the best possible consistency for the various applications of these mortars (plastering, coating, laying brickwork, facade regularization, etc.) [82].

In this work, the consistency of the tested mortars was determined using a flow table keeping with the standard EN 1015-6 [83]. The parameters determined in this test are the flow values, which were then related to the WBA disposition and proportion.

The flow values for the mortars tested are summarized in Table 11, along with the density, air content, and temperature of the mortar immediately after mixing.

To maintain a constant w/b ratio while achieving the recommended flow diameter of 162 to 168 mm [84], the dosage of polycarboxylate superplasticizer varied from 0.11% in LM0 to 1.0% in LM1-20. It was found that a lower amount of superplasticizer was enforced by WBA3, while the superplasticizer requirement increased with WBA content, suggesting a similar behavioral tendency in which the water demand increases with the amount of WBA. In the mortar mixes with WBA1, the upper limit of 1% was reached in the

proportioning of superplasticizer; therefore, the lowest flow value of 150 mm was kept in order not to impair the mechanical properties too much.

**Table 11.** Fresh properties of AHL mortars.

| Lime Mortar ID | LM0 | LM1-20 | LM1-30 | LM3-20 | LM3-30 |
|---|---|---|---|---|---|
| Air-entraining agent (wt.%) | | | 0.02 | | |
| Polycarboxylate super p. (wt.%) | 0.11 | 1.0 | 1.0 | 0.44 | 0.84 |
| w/b ratio | | | 0.60 | | |
| Bulk density (kg/m$^3$) | 2055.2 | 2040.2 | 2055.2 | 2046.3 | 1996.1 |
| Air content (%) | 7 | 7.6 | 6 | 8 | 9.4 |
| Flow (mm) | 161 | 158 | 150 | 161 | 167 |
| Temperature (°C) | 23.8 | 24.2 | 25.3 | 25 | 25.1 |

All of the lime mortar mixes exhibited similar workability regardless of composition, with no significant segregation or bleeding noted after testing. Nevertheless, the most comparable flow values were measured for mixes with a lower WBA content, i.e., LM1-20 and LM3-20, but the diameter of 167 mm measured for LM3-30 was also within the desired range, which should ensure good workability on a construction site.

Based on previous studies [75,85], it was assumed that the addition of biomass fly ash would increase the required amount of air-entraining agent and at the same time may lead to instability of the air content; therefore, a relatively high proportion was tentatively chosen for all mixes. Although the air-entraining agent was added in equal amounts in all the mixes, the air content varied from the lowest value of 6% in LM1-30 to the highest value of 9.4% in LM3-30. Slightly elevated temperatures were observed for all mortars containing WBA compared to the reference mix, and this was especially true for mixes with higher proportions of WBA. A maximum temperature increase of up to 6 °C was observed for the LM1-30 and LM3-30 mixes. Although it is acknowledged that higher temperatures lead to faster setting [18], this trend was not observed in the AHL pastes with WBA, while setting was extremely delayed in all AHL pastes and increased with the WBA content.

3.3.2. Mechanical Properties

The influence of each WBA on the mechanical properties of the AHL mortar was evaluated based on the demonstrated compressive and flexural strength. The average values of compressive and flexural strengths of mortars cured for 28 days under various conditions (HC, DC, ACC) and the calculated values of SAI are summarized in Table 12, along with the ratio of compressive to flexural strength $f_C/f_F$ categorized by the NHL content of the binder blends. The reported ratio of compressive to flexural strength was used to evaluate the potential ductility of the AHL mortars [86].

In addition to the above properties, compressive strength has been recognized as a starting point for evaluating the performance of a hardened mortar. From the absolute values of compressive strength (see Figure 5), it can be recognized that the addition of both WBA1 and WBA3 to the binder system NHL-WBA contributes to the increase in compressive strength of AHL mortars in both WBA hybridization ratios. Mortars with a higher degree of hybridization of 30 wt.% showed comparable or even higher values for compressive strength, while the results for flexural strength followed the same trend. The positive contribution is present in all three curing regimes, while the highest values of compressive strength were obtained under the conditions of ACC. Enhanced carbonation improved the strength of AHL mortars with WBA by up to 18% (LM3-30) compared to the reference mix. In the comparison between natural and enhanced carbonation, the reference mortar LM0 showed a significant increase in compressive strength of about 63% when exposed to a high $CO_2$ concentration. Regardless of the staggered contribution in mixes with WBA, a positive change in ACC conditions is evident. Seeing that the compressive strengths of the mortars with WBA were comparable to those of the reference mix after being

exposed to ACC, it can be assumed that the rather higher values obtained at DC are due to the synergistic reaction of lime and WBA themselves, in addition to natural carbonation, since it is assumed that most of the strength was obtained by pozzolanic properties and not by simulating aging. The fine particles from WBA may also have contributed to the potential pozzolanic activity, which has also been investigated in previous studies [87]. On average, NHL repair mortars can reach about 50% of their final strength during the first 28 days of curing [9]. For this reason, it can be concluded that the values achieved under the ACC conditions are not far from the estimated final values. Interpreting the results of the compressive strength of mortars subjected to HC, which was expected to be most favorable for this type of material [11,45,51,88], it is clear that lower values were obtained than in DC, with the exception of the reference mortar, which showed better results in a humid environment. Thus, moderate moisture conditions, such as those maintained in the DC regime, can ensure the higher mechanical properties of AHL mortars, which is consistent with [51]. Conforming to ASTM C618 [11], a strength activity index greater than 75% after 28 days indicates pozzolanic reactivity when the cementitious binder is replaced with 20% coal fly ash. According to EN 450-1 [89], achieving a result of more than 75% with a 25% substitution is considered a confirmation of pozzolanic reactivity after 28 days. Since the calculated SAI values were obtained using wood ash as the NHL substitute and at an even higher degree of substitution, the values of SAI were used in this study for comparison purposes only and not for acceptance under these standards. However, the calculated SAI was higher for all AHL mortars with WBA than for the reference mix, which was especially highlighted for the mortars exposed to DC, corroborating the pozzolanity of WBA.

**Table 12.** Compressive and flexural strength of mortar after 28 days under different curing conditions.

| Mix ID | NHL-Content in Binder Blends (%) | Compressive Strength, $f_C$ (MPa) | Flexural Strength, $f_F$ (MPa) | Compressive to Flexural Strength Ratio $f_C/f_F$ | SAI (%) |
|---|---|---|---|---|---|
| | | **HC conditions** | | | |
| **LM0** | 100 | 4.22 (±0.06) | 1.27 (±0.08) | 3.31 | - |
| **LM1-20** | 80 | 5.45 (±0.12) | 2.08 (±0.14) | 2.61 | 129.15 |
| **LM3-20** | | 5.56 (±0.12) | 1.99 (±0.03) | 2.80 | 131.75 |
| **LM1-30** | 70 | 6.32 (±0.38) | 2.70 (±0.10) | 2.34 | 149.76 |
| **LM3-30** | | 5.99 (±0.16) | 2.08 (±0.12) | 2.88 | 141.94 |
| | | **DC conditions** | | | |
| **LM0** | 100 | 3.47 (±0.26) | 0.96 (±0.04) | 3.61 | - |
| **LM1-20** | 80 | 6.23 (±0.23) | 1.77 (±0.23) | 3.53 | 179.54 |
| **LM3-20** | | 7.11 (±0.12) | 2.10 (±0.03) | 3.39 | 204.89 |
| **LM1-30** | 70 | 6.80 (±0.11) | 1.68 (±0.04) | 4.05 | 195.97 |
| **LM3-30** | | 6.62 (±0.05) | 2.26 (±0.01) | 2.93 | 190.78 |
| | | **ACC conditions** | | | |
| **LM0** | 100 | 9.29 (±0.09) | 2.19 (±0.08) | 4.23 | - |
| **LM1-20** | 80 | 10.09 (±0.45) | 2.84 (±0.14) | 3.55 | 108.73 |
| **LM3-20** | | 10.45 (±0.55) | 2.92 (±0.14) | 3.58 | 112.61 |
| **LM1-30** | 70 | 9.31 (±0.32) | 1.95 (±0.04) | 4.78 | 100.32 |
| **LM3-30** | | 10.98 (±0.16) | 2.60 (±0.32) | 4.22 | 118.19 |

(the numbers in brackets are standard deviations).

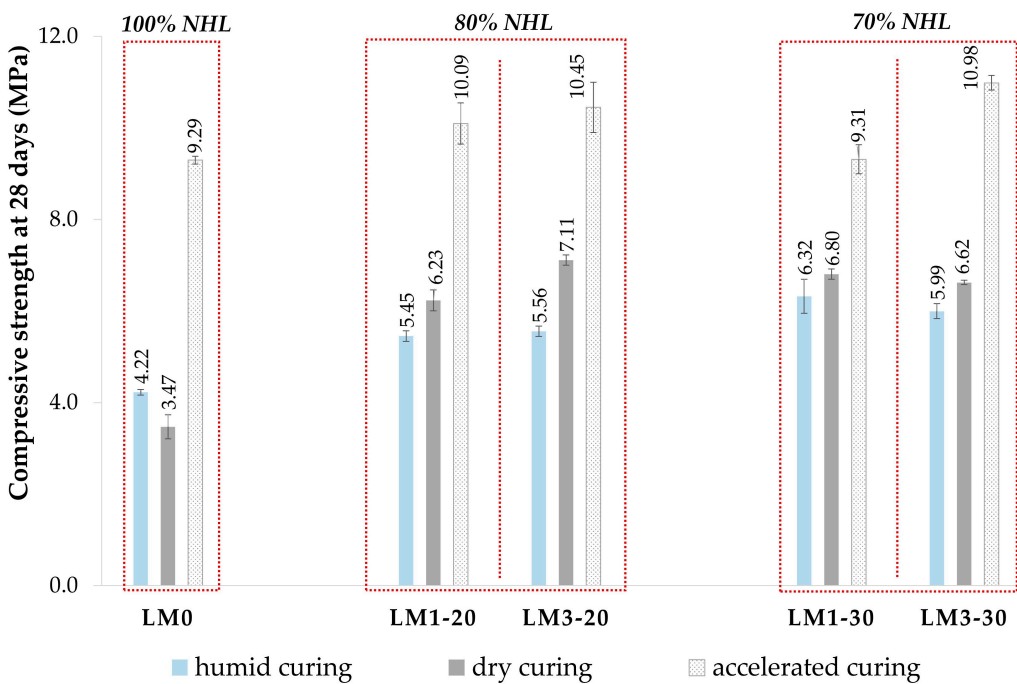

**Figure 5.** Compressive strength of mortars after 28 days of different curing conditions.

An increased strength of repair mortars may indicate the improved durability of the mortar, but at the same time, an excessive increase may be detrimental to the surrounding masonry. If the mortar is too stiff, dense, and impermeable, stress concentrations may occur (e.g., strong repointing mortars may cause damage to adjacent masonry). Care should therefore be taken to ensure that the compressive strength is not disproportionately high. The compressive strength limit for repair mortars is usually between 6 and 10 MPa for mortars used in older, historic masonry [53].

Several studies [86,90,91] suggest that the relation between compressive and flexural strength is proportional to the modulus of elasticity in tests lasting from 1 to 6 months, assuming that a low ratio between compressive and flexural strength ($f_C/f_F$) correlates with a low modulus of elasticity. Similarly, the lower the modulus of elasticity values and the lower the $f_C/f_F$ ratio, the higher the flexibility of the mortar is likely to be. Therefore, mortars with a low $f_C/f_F$ ratio are potentially equivalent to authentic materials due to their elastic behavior, thus ensuring a higher durability. Accordingly, the $f_C/f_F$ ratio is emphasized as a simplified method for evaluating the potential ductility of mortars. The values of the $f_C/f_F$ ratio for the AHL mortars with only NHL as binder and the AHL mortars with WBA (Table 12) are consistent with the ranges previously reported in the literature, where the tested mortars exhibit elastic behavior, compatible with that of historic mortars. While the curing conditions do not seem to have too much influence on the flexural strength, the $f_C/f_F$ values were highest in the DC and ACC environments and also lower than the reference values, with the exception of LM1-30.

Based on all the measurements of the carbonation depth (Table 13), it can be seen that the mortars with WBA up to 28 days in DC tend to carbonate somewhat slower, most clearly in the LM1 mortars. Due to the lower porosity and higher superplasticizer content in the AHL mortar mixes with WBA, carbonation may have been hindered, while the admixture may also have had an inhibitory effect on the precipitation of $CaCO_3$ [47]. Just as it was found that a high relative humidity of up to 95% prevented carbonation, the samples were fully carbonated after 28 days of ACC.

**Table 13.** Evaluation of carbonation depth on AHL mortar halves (phenolphthalein staining).

| Mortar Mix ID | HC | | DC | | ACC |
| --- | --- | --- | --- | --- | --- |
| LM0 | | 44% | | | |
| LM1-20 | | 32% | | | |
| LM3-20 | up to 5% | 29% | | 100% | |
| LM1-30 | | 39% | | | |
| LM3-30 | | 37% | | | |

### 3.3.3. Pore Structure

The porosity parameters of the tested mortars, such as total permeable porosity and critical pore radius, are profiled in Table 14 and Figure 6, wherein the fully non-carbonated (NCR) and carbonated (CR) specimens were analyzed using MIP, after curing in HC and ACC, respectively. Only the higher hybridization level of 30% WBA was assessed due to its reported enhanced mechanical properties while being more competent in the context of sustainability. The NHL reference mortar (LM0) proved to be one of the most porous mortars, in contrast to the AHL mortars with 30 wt.% WBA, which showed slightly reduced porosity values. This is even more evident in the non-carbonated specimens with WBA in relation to the reference mortar, where the decrease in permeable porosity is 2.12% and 1.24% for LM1-30-NCR and LM3-30-NCR, respectively. For the carbonated specimens, a decrease is observed only for LM1-30-CR with 1.08%, while the total porosity of LM3-30-CR is in line with the carbonated reference mortar.

The formation of calcium silicate hydrates (C-S-H) in hydraulic limes is generally associated with a reduction in porosity due to the growth of hydrated structures [92]. In consideration of the increased compressive strength of mortars with WBA, as well as the reduced permeable porosity, their pozzolanic reaction with portlandite from lime could lead to additional filling of the pores with C-S-H. The hydration reactions in NHL may be upgraded by adding pozzolanic material, initializing a pozzolanic reaction with water and a carbonization reaction continuously occurring in pozzolanic material–NHL-based binders. At the same time, the non-carbonated $Ca(OH)_2$ dehydrates and recrystallizes in a dry environment to form a compact structure with C-S-H [8].

**Table 14.** Porosity parameters at 28 days of mortars cured in HC and ACC obtained via mercury intrusion porosimetry (MIP).

| | | Critical Pore Entry Radius | Median Pore Diameter | | Average Pore Diameter | Permeable Porosity |
|---|---|---|---|---|---|---|
| | | | Volume | Area | | |
| | Curing Conditions | μm | μm | μm | μm | % |
| **HC** | LM0-NCR | 0.84 | 0.554 | 0.016 | 0.059 | 28.75 |
| | LM1-30-NCR | 0.05 | 0.052 | 0.027 | 0.041 | 26.63 |
| | LM3-30-NCR | 0.06 | 0.075 | 0.026 | 0.048 | 27.51 |
| **ACC** | LM0-CR | 0.68 | 0.435 | 0.024 | 0.091 | 25.19 |
| | LM1-30-CR | 0.15 | 0.156 | 0.023 | 0.065 | 24.11 |
| | LM3-30-CR | 0.55 | 0.297 | 0.024 | 0.081 | 25.44 |

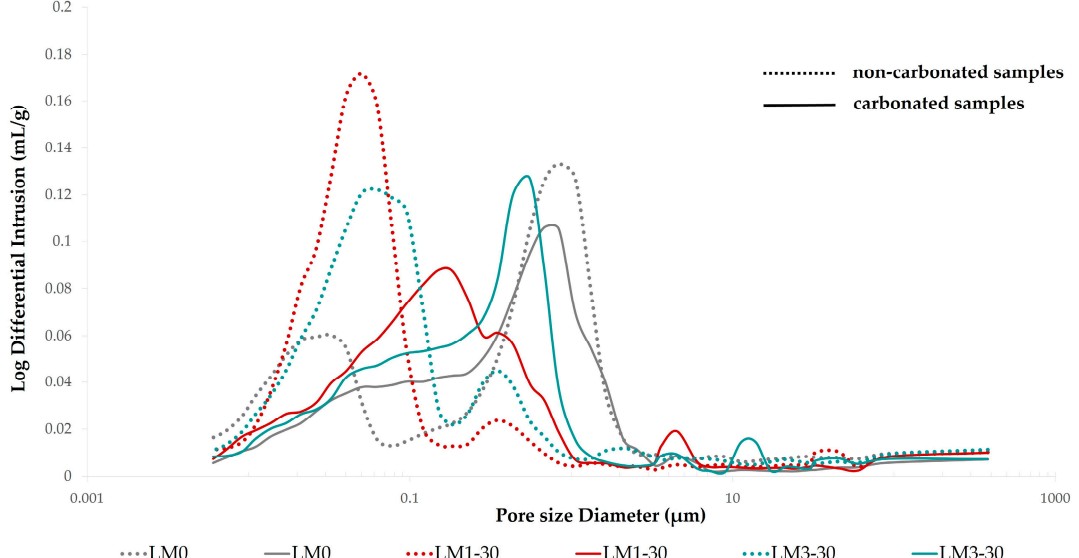

**Figure 6.** Pore size distribution in non-carbonated and carbonated mortar specimen.

While the critical pore entry radius illustrates the grouping of the largest fraction of interconnected pores in the binder system [92], it corresponds to the peak in the differential pore volume curve, implying the pore size corresponding to the maximum volume intrusion. All values of the critical pore entry radius are listed in Table 14, with a sharp increase for the AHL mortars with WBA in response to carbonation. While LM0 had the highest value of critical pore entry radius (0.84 μm) under HC conditions, which tended to decrease with carbonation, the critical pore entry radii of LM1-30 and LM3-30 are significantly lower, which can also be seen on the differential pore volume curve (Figure 6), where a shift toward higher pore radii is evident in the case of LM0. The porosity of the carbonated structure has evolved such that the LM3-30 mortar has similar peak values to the reference mortar, while the slightly lower peak value of LM1-30 is positioned somewhere between the NCR and CR values.

Figure 6 shows the pore size distribution for non-carbonated and carbonated AHL mortar specimens. As can be seen, the non-carbonated specimens exhibit a bimodal distribution with two sharp peaks of the main volume intruded, while the carbonated specimen showed a unimodal distribution. In particular, the non-carbonated mortar specimen of LM1-30 and LM3-30 showed a maximum of the intruded volume for pores around 0.05 μm, while for the LM0 mortar, the main peak was shifted to about 0.8 μm, and the secondary peak occurred around 0.03 um. Thus, compared to the reference mortar

mix, the AHL mortars with WBA had a higher number of pores with a diameter less than 0.1 μm and a much lower number of pores with a diameter greater than 1 μm, as disclosed in Figure 6. Therefore, a shift of the peaks related to small/medium pores to higher pore diameters is visible in the carbonated specimen of AHL mortars with WBA. As mentioned above, the unimodal distribution showed a maximum of the intruded volume for pores around 0.2 μm (LM1-30) and 0.5 (LM3-30). After carbonation, the reference mortar had a lower intruded volume at about 0.8 μm. It is also noted that the carbonated AHL mortars resulted in curves with a slightly wider pore range, reduced intrusion peaks, and a higher percentage of smaller capillary pores in the range of 0.1–1 um [87].

In this paper, the relative pore volume distribution (%) of the investigated AHL mortars is determined for specific pore radius ranges (Table 15). For the selection of the pore radii, the ranges proposed by other researchers for lime mortars were taken into account [93–95].

**Table 15.** Distribution of relative pore volume (%) in different ranges of pore radius.

| Ranges of Pore Radius | NCR | | | CR | | |
|---|---|---|---|---|---|---|
| | LM0 | LM1-30 | LM3-30 | LM0 | LM1-30 | LM3-30 |
| **<0.01 μm** | 4% | 3% | 3% | 2% | 3% | 2% |
| **0.01–0.05 μm** | 23% | 46% | 32% | 14% | 18% | 15% |
| **0.05–0.1 μm** | 3% | 27% | 24% | 9% | 15% | 11% |
| **0.1–0.5 μm** | 20% | 11% | 23% | 33% | 45% | 45% |
| **0.5–1 μm** | 24% | 2% | 3% | 23% | 6% | 14% |
| **>1 μm** | 26% | 12% | 15% | 20% | 14% | 13% |
| **Total** | 100% | 100% | 100% | 100% | 100% | 100% |

Pores with a radius of less than 1 μm are considered to be connected to the binder [96] and represent the majority in the AHL mortars studied. Pores outside this range are present for the most part in the reference mortar LM0 with 26% for HC and 20 for ACC conditions. As for the AHL mortars with WBA, pores with a radius of more than 1.0 μm are represented by up to 15%, without a correlation with the WBA content being discernible. The reference LM0 mortar appears to have a slightly greater proportion of gel pores (pore radii < 0.01 um) than the LM1-30 and LM3-30 mortars, which are strongly associated with the formation of hydration products [93]. This could be due to the fact that 30 wt.% of NHL 3.5 was replaced by WBA as a secondary binder component, thus affecting the hydraulics of the binder blends. Likewise, the proportion of gel pores decreases from 4% in the non-carbonated LM0 to 2% in the carbonated specimen, where hydration reactions are put off or even terminated by the complete carbonation of lime.

A high relative pore volume in the pore range of 0.1–1 μm is often associated with lime-based mortars [87], and a number of authors have linked pores of this size with carbonation and hydration reactions, while pores with a radius of about 0.5 um are associated with the conversion of portlandite to calcite [87,93]. All the mortars investigated have high porosity in this pore range, especially notable in the carbonated specimen (over 50% in all the mixes), which indicates promising compatibility with historic mortars, which have the highest pore volume in the pore range of >0.1 μm [93]. In addition, the pore volume associated with the above-mentioned pore radius of 0.5 um, i.e., the pore range of 0.1–0.5 μm, showed significant increases of 13%, 34%, and 22%, respectively, for the carbonated specimen from the LM0, LM1-30, and LM3-30 mixes, which is consistent with the fully carbonated structure. While this is the predominant pore range in all carbonated specimens and reveals larger pores in the carbonated structure of the mortars, the range of 0.01–0.05 μm is the prevailing one in the still non-carbonated structure. In this so-called mesopore region, defined by a pore diameter of less than 0.5 μm, the mortars with WBA cured in HC exhibit

a significant pore concentration, presumably due to the hydration of reactive amorphous compounds [92]. In the non-carbonated sample of LM1-30, this pore range accounts for 46%, while it doubled compared to the LM0 mix, and in LM3-30, it was 32%, corresponding to an increase of 1/3.

### 3.3.4. Mineralogical and Morphological Features of AHL Mortars

To further investigate the changes in the microstructure of the mortars after exposure to different $CO_2$ and moisture conditions, samples of the early aged AHL mortars were analyzed using XRD. The HC and ACC environments were considered to shed light on the changes after complete carbonation.

As shown in Figure 7, the main phases in all the AHL mortars studied are quartz (highlighted in yellow), calcium carbonate, and portlandite, with a coherent difference between fully carbonated (ACC) and non-carbonated samples (HC), in terms of both WBAs concerned. Portlandite is thus present in LM1-30 and LM3-30 when exposed to HC, while in the carbonated samples, it converts to calcium carbonate as the characteristic peaks are no longer visible. That this reaction has occurred is also confirmed by the increase in intensity of the main calcite peak (marked in grey) and the disappearance of the peaks associated with portlandite (marked in blue). The diffractograms are relative, i.e., the change in intensity is a relevant parameter for the change in the contribution of a single component to the mortar composition. It is interesting to note that the samples cured in HC show several peaks below 20 degrees. These peaks at low angles around 5 to 15 usually indicate some kind of hydrate in such systems, in this case possibly C-S-H phases, accompanied by the very humid atmosphere in which the mortars were cured. The peaks mentioned in the range of 5 to 15 degrees are, like portlandite, sensitive to carbonation. The reason these peaks do not appear in the carbonated samples below 20 degrees is that the C-S-H phase reacts with $CO_2$ and shifts the chemical reaction to the additional formation of calcite, $CaCO_3$. Accordingly, the peaks are of weak intensity, but they are present in the samples cured in HC, while they are not present in the samples exposed to ACC.

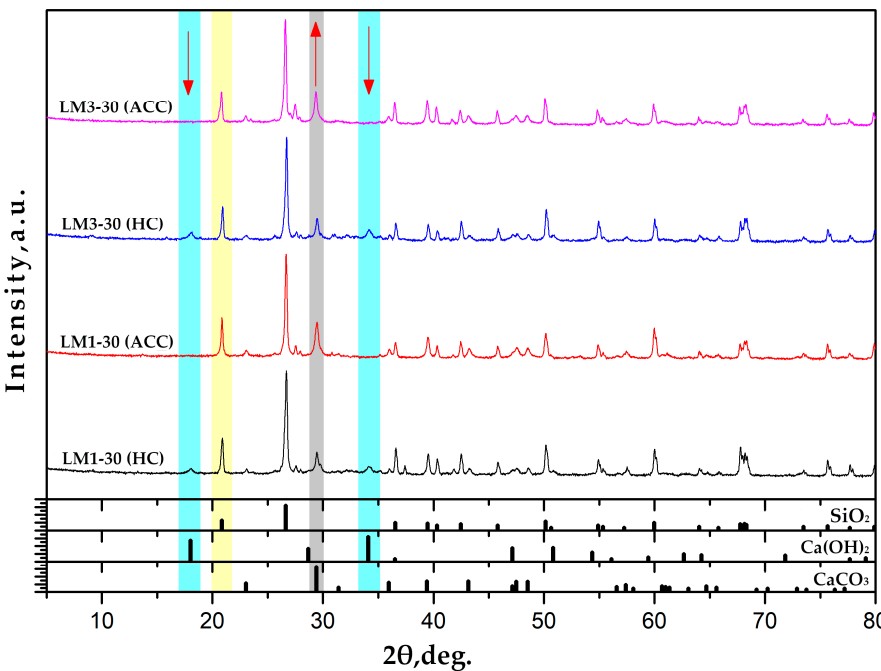

**Figure 7.** XRD diffractograms of mortars with WBA cured for 28 days in ACC and HC conditions.

The microscopic morphology of the AHL mortars was analyzed using SEM as shown in Figure 8, where Figure 8a,c,e displays the non-carbonated structures and Figure 8b,d,f shows the carbonated structures, which are shown as 3500× magnified micrographs. After

28 days of HC, there were just a few portlandite crystals visible in the reference mortar LM0, dispersed among the numerous needle-like hydration products with calcite crystals (Figure 8a), along with pores with a rounded morphology. Assuming that the formation of secondary C-S-H results from the pozzolanic reaction between WBA and $Ca(OH)_2$, a more homogeneous and compact microstructure is visible in the SEM micrographs of LM1-30 and LM3-30. The SEM images of the carbonated matrixes of all AHL mortars analyzed, accentuated by forced carbonation, show calcite crystals with an abundance of sand (Figure 8b,d,f). In terms of morphology, the calcite crystals in the mortar samples cured in a $CO_2$-enriched environment were apparently somewhat larger and better developed, forming a more interlocked and denser structure, which certainly contributed to the higher strengths of these mortars. According to De Silva et al. [47], the leading factor affecting strength is not the amount of $Ca(OH)_2$ converted, but the morphology of the $CaCO_3$ crystals formed. Well-developed crystalline structures and crystal habits lead to better binder properties.

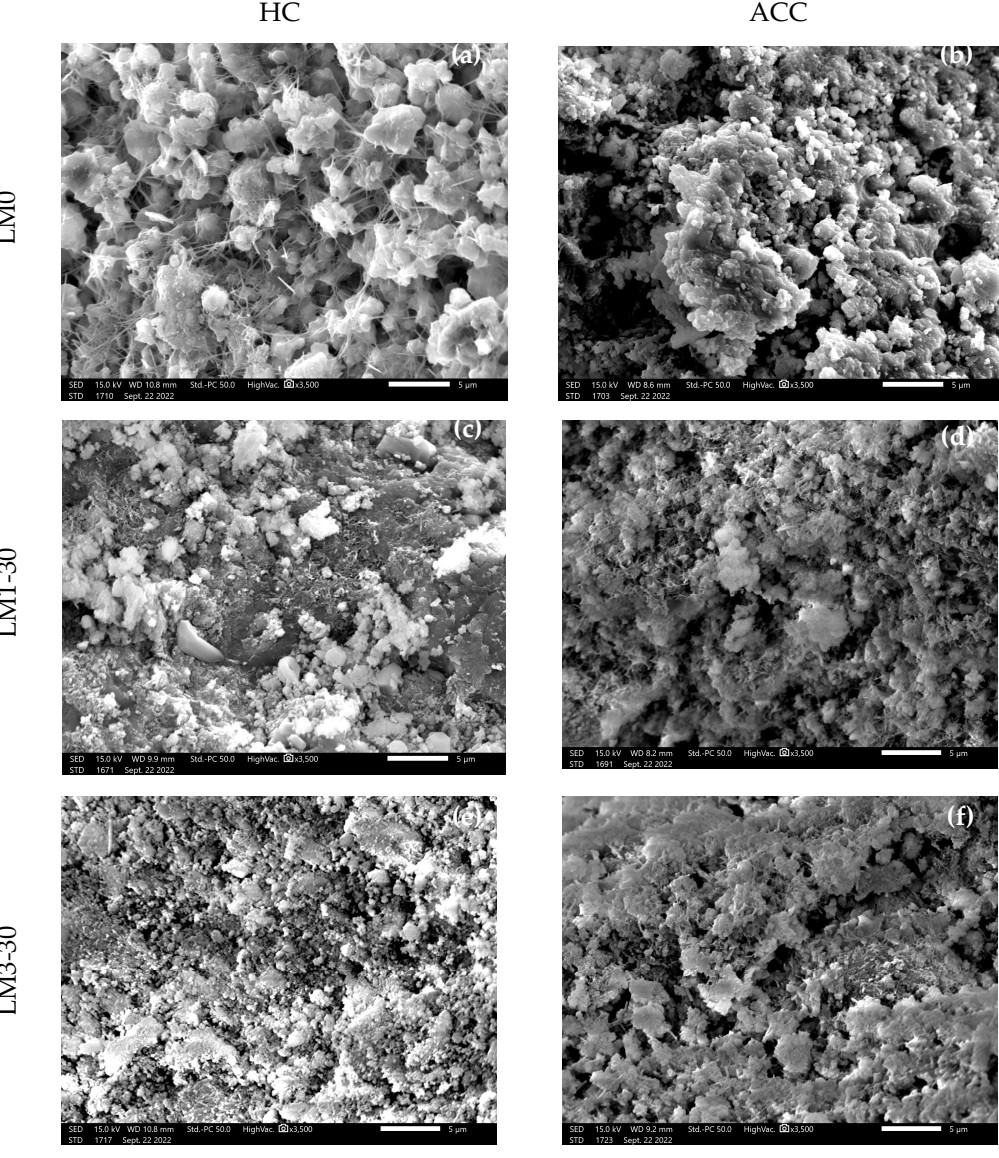

**Figure 8.** SEM microphotographs of AHL mortars at HC (**a**,**c**,**e**) and ACC (**b**,**d**,**f**) at 3500× magnification.

## 4. Conclusions

Experimental analysis of the basic physical, chemical, and mineralogical attributes of hybrid lime binders modified by WBA revealed bright prospects for the integration of WBA into lime mortars for conservation purposes.

Just as fine particles ($<50$ µm) dominate the particle size distribution in WBAs, their chemical composition with a dominant proportion of $CaO$, $SiO_2$, and $Al_2O_3$ provided a suitable framework for improving the hydraulicity of traditional binders such as NHL. Both CI and HI values indicate that the AHL blends exhibit high hydraulic performance comparable to that of natural cement.

However, the high $SO_3$ content together with the high phosphate and alkali content disclosed in both WBAs affects the initial and final strength of the AHL pastes. The strong hindering effect of WBA on the setting of AHL blends was also attributed to high heavy metal concentrations, which underlines the fact that the delayed setting of AHL pastes is strongly governed by the chemical composition of the ashes.

On the other hand, the incorporation of WBA as a low-carbon secondary binder into the NHL system in the two NHL/WBA hybridization ratios of 80:20 and 70:30 revealed an overall positive effect on the mechanical properties, indicating pozzolanic activity. All AHL mortars could be declared as general purpose masonry mortar (G) within the M5 mortar classification (in terms of compressive strength), which falls within the range specified by EN 998-2 [97].

Semi-dry conditions with a relative humidity up to 70% with access of air seemed to be the most favorable for the carbonation and hardening of AHL mortars, enhancing their mechanical properties. All the mortars investigated show high porosity in the pore range of 0.1–1 µm, particularly noticeable in the carbonated structures, indicating promising compatibility with historic mortars.

Thus, the results obtained show that fly WBA could be successfully utilized in a moderate proportion (up to 30% by weight), which allows the development of an environmentally friendly lime mortar suitable for conservation purposes.

In the context of efforts to address SDG #12, responsible consumption and production, the genuine goal of zero waste is not only to avoid landfilling waste, but to reshape the entire cycle of resource extraction, utilization, and waste management.

Thus, the integration of waste materials, such as WBA, within the existing historical framework, employing a closed-loop design, has unequivocally established itself as a groundbreaking paradigm shift. This shift is imperative for promoting long-term, holistic thinking within the industrial ecosystem, considering the entire life cycle of products and processes.

**Author Contributions:** Conceptualization, J.Š.B., N.Š. and A.B.; methodology, J.Š.B. and A.B.; validation, N.Š.; formal analysis, J.Š.B., N.Š. and A.B.; investigation, J.Š.B.; resources, N.Š.; data curation, J.Š.B.; writing—original draft preparation, J.Š.B.; writing—review and editing, N.Š.; visualization, J.Š.B.; supervision, N.Š.; project administration, J.Š.B.; funding acquisition, N.Š. All authors have read and agreed to the published version of the manuscript.

**Funding:** We wish to acknowledge the support received from the Croatian Science Foundation and the financial backing provided through the European Social Fund under the project entitled "Young Researchers' Career Development Project–Training New Doctoral Students" (ESF DOK-01-2018).

**Institutional Review Board Statement:** Not applicable.

**Informed Consent Statement:** Not applicable.

**Data Availability Statement:** Not applicable.

**Acknowledgments:** Some of the results presented were obtained using equipment provided by scientific projects "Alternative binders for concrete: understanding the microstructure to predict durability, ABC" (UIP-05-2017-4767), "Development of new innovative project ECO$_2$ Flex" (EFRR IRI II-KK.01.2.1.02.004) and "Development of innovative building composites using bioash" (KK.01.2.1.01.0049).

**Conflicts of Interest:** The authors declare no conflict of interest.

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
