# Peer review of "Sustainable Hybrid Lime Mortars for Historic Building Conservation: Incorporating Wood Biomass Ash as a Low-Carbon Secondary Binder"

_heritage, doi:10.3390/heritage6070278_

Round 1

Reviewer 1 Report

The manuscript deals with a subject worthy of investigation, encompassing two well-known constituents: natural hydraulic lime and wood biomass ashes.

If the NFL does not raise major questions about its use in conservation actions and restoration of built heritage, wood biomass ashes are more questionable due to their chemical characteristics (high levels of alkalis and sulfates). The authors assumed that the application of these ashes at levels up to 30% would not imply problems in terms of durability, but they did not carry out tests that showed that these problems could not arise, namely the formation of salts or deleterious reactions with the aggregates.

In any case, the article presents a set of results that deserve to be published. However, before that, the authors must clarify a set of questions essentially related to the characterization of the materials they used, which I pointed out in the PDF of the submitted manuscript.

Therefore, my opinion is that the manuscript is conditionally accepted, depending on the implementation of the corrections indicated in the revised PDF.

Please see my suggestions in the attached revised manuscript.

Reviewer 2 Report

The authors of the presented paper worked on the Sustainable Hybrid Lime Mortars for Historic Building Conservation. The topic is interesting; however, certain minor changes are required in the manuscript to be in acceptable form. The specific comments are:

·       The authors gave a weak abstract. Utilization of the study is missing which needs to be added.

·       The introduction part can be improved by adding more recent and relevant literature.

·       Why the authors specifically selected hydraulic lime as the primary binder and wood biomass fly ash (WBA) as the secondary substitute; its justification is missing.

·    The English language and formatting need to be improved throughout the manuscript.

·       The graphical quality of Figure 2 and 3 can be enhanced.

·       The results and discussion part can be improved by comparing it with the previously conducted studies.

·       The English language and formatting need to be improved throughout the manuscript.

Reviewer 3 Report

It is suggested that the introduction be revised. The complexity of the research and the quantity of studies carried out necessitate a lengthy discussion, so the long introduction can be reduced in favour of the readability of the paper

page 12 - 392 line, the data from the thermal analysis have already been mentioned and need not be repeated

Round 2

Reviewer 1 Report

Most of my suggested revisions were taken into account in this version 2 of the article. However, as listed below, some small details are yet to be verified by the authors, some not previously referred to in my first review:

In Table 1 there is a blank line that must be eliminated.

Lines 227 and 228:  The Rietveld method is used to obtain the semiquantitative composition, and not the qualitative composition. This sentence should be modified. The test conditions used in the XRD analysis need to be improved (voltage, intensity, type of X-ray radiation, step size, …), as well as more information about the semi-quantitative XRD analysis. What PDF files and COD structures are used? What are the refinement parameters adopted? If possible, include at least one example of the refinement of the samples studied.

Lines 257-263: The test conditions used in SEM analysis are incomplete; the voltage and current used are missing. The SEM was probably accompanied by the EDS analysis, which if confirmed should also be mentioned.

Table 6: The chemical compositions should be corrected. The sums greatly exceed the total of 100% (e.g., in NHL 3.5 sample the sum is 116,12%). Also, the mineralogical composition does not reflect the application of the Rietveld method, as mentioned previously by the authors in lines 227 and 228. Furthermore, the authors refer to the presence of γ-belite in WB samples, which is surprising and must be commented.

Lines 354-361: The description of NHL 3.5 composition does not match the minerals present in table 6. Furthermore, the authors do not comment on which minerals could explain the high K2O values of the WB samples.

Figure 3 and lines 414-425: The sample WBA1 is the only one that shows two peaks in decarbonation zone. This deserves to be commented on. When dolomite is present, this effect occurs.
